# The Role of the Electron Transport Layer in the Degradation of Organic Photovoltaic Cells

Alaa Al-Ahmad 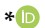, Benjamin Vaughan, John Holdsworth, Warwick Belcher, Xiaojing Zhou * and Paul Dastoor *

Center of Organic Electronics, University of Newcastle, Newcastle 2308, Australia;
alaa.alahmad@newcastle.edu.au (A.A.-A.); ben.vaughan@newcastle.edu.au (B.V.);
john.holdsworth@newcastle.edu.au (J.H.); warwick.belcher@newcastle.edu.au (W.B.)
* Correspondence: xiaojing.zhou@newcastle.edu.au (X.Z.); paul.dastoor@newcastle.edu.au (P.D.)

**Abstract:** The performance of the electron transport layer (ETL) plays a critical role in extending the operational lifespan of organic photovoltaic devices. ZnO is an excellent electron transport layer used in the printable organic photovoltaic cells. A comparison of Ca and ZnO as the ETL in encapsulated bulk heterojunction OPV devices has been undertaken with the device stability dependence on light soaking, temperature, irradiance, and thermal cycling recorded. It was observed that the OPV devices using Ca ETL decayed faster than the ZnO ETL devices under the same light illumination. The degradation in a Ca ETL device is ascribed to the formation of an insulating calcium oxide layer at the ETL interfaces. Photoluminescence (PL) spectroscopy revealed a higher PL signal for the degraded Ca ETL devices compared to the ZnO ETL devices. Power conversion efficiency (PCE) of the ZnO ETL devices was found to be much more stable than the Ca devices. The PCE for ZnO ETL devices still retained 40% of their initial value while the Ca ETL devices failed completely over the period of 18 days in the study, leading to a clear outcome of the study.

**Keywords:** electron transport layer; light soaking; OPV degradation

## 1. Introduction

Solar energy from photovoltaic cells is a most promising source of renewable energy with a distinct advantage compared to other renewable sources, being that the Sun's radiation can be directly converted into electrical energy without any need for mechanical parts. Each day the Sun provides 10,000 times the energy needed on the planet [1–3]. The International Energy Agency (IEA) has estimated that by 2030, renewable energy will contribute 450 billion kWh per year [4]. Today, a small but increasing fraction of power is generated by solar cells since they remain expensive compared to the price of power from a fossil fuel-based generation and grid system, but their advantage is that they are an environmentally friendly, non-polluting, and low maintenance energy source. The growth of such technology depends on its competitiveness, considering factors such as material efficiency and availability, cost-effectiveness, reliability, and structure development [5]; however, the goal will always be maximum energy at minimum cost [6,7].

Solar cells based on organic polymers and small organic molecules are under developmental research with the aim to lower the PV cell cost [8]. Recently, a 16.5% conversion efficiency of an organic cell and a 17.3% efficiency tandem organic cell were reported [9–13]. The economic viability calculation highlights that organic photovoltaic (OPV) production becomes competitive with traditional photovoltaic (PV) technologies [14] provided a device efficiency greater than 3% and a module lifetime exceeding 5 years can be achieved [15,16].

Organic materials used in OPV devices are by nature more susceptible to both chemical and physical degradation [17,18], therefore, the device performance is not constant over time and has been found to depend strongly on environmental factors such as illumination, temperature, and humidity [19]. The degradation of OPV devices and materials is a

complex problem, not yet entirely understood [20], and remains a key factor in OPV device commercialization. Understanding and solving the present limits to OPV device lifetime is a priority [21] with research focused on understanding long-term OPV device degradation mechanisms underway in many laboratories [22]. The degradation mechanisms affect the active layer, interfaces, holes/electron transport layers, and contacts of OPV devices [23]. Interestingly, Saeed et al. studied OPV using the indoor light conditions, the degradation processes were probed, and solutions were proposed [24,25].

Mateker and McGehee published an excellent review on OPV degradation [26] and pointed out that it is essential to encapsulate the devices to significantly reduce extrinsic degradation and necessary for long-term stability. Even with inverting the polarity of OPV devices, it cannot prevent material bleaching under illumination. More recently, Kim and co-workers proposed that increasing the active components from two to four to promote a more stable morphology in the active layer at the molecular level and improves the devices stability [27].

Here we report a systematic study on encapsulated OPV with a standard polarity of electrodes, in particular to examine the degradation due to the electron transport layers (ETL), either calcium (vacuum evaporated) or ZnO (spin-coated). Factors that have been found to affect OPV degradation, including light soaking (continuous illumination), temperature, irradiance, and thermal cycling, were carefully studied. Furthermore, the light illumination protocol and the dark thermal aging combination were carefully probed to isolate the effects of photo-induced degradation, as well as to mimic the similar degradation process that OPV devices experience in real-world operation. ZnO, which a printable layer, was identified and demonstrated as a more superior ETL for the OPV devices in this study.

## 2. Experimental Section

Two OPV devices with different electron transport layer were produced through spin coating active media onto transparent electrodes as shown schematically in Figure 1. These were subjected to laboratory weathering and dark storage degradation protocols for long-term continuous illumination under the 100 mW/cm$^2$ irradiance.

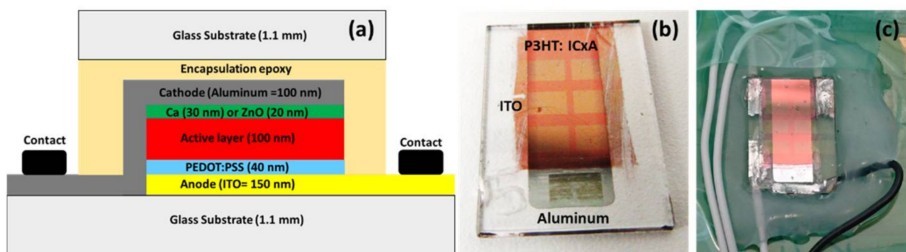

**Figure 1.** (**a**) The schematic structure of a standard OPV device using either Ca or ZnO as the electron transport layer. (**b**) An actual P3HT:ICxA organic photovoltaic device (active layer thickness 100 nm) prepared on XY15S indium tin oxide (ITO-) glass substrate. (**c**) A fully encapsulated P3HT:ICxA device with wires soldered to the ITO and aluminum contacts.

### 2.1. Materials

All supplied materials were used without additional purification unless otherwise stated. Patterned ITO slides (15 Ohm/sq) with a size 12.5 mm × 17.5 mm were purchased from Xinyan Technology Ltd., Nanjing, China. Poly (3,4-ethylene dioxythiophene): polystyrene sulfonate (PEDOT:PSS) (PVP Al 4083, Clevios) was purchased from Heraeus, Hanau, Germany. Poly(3-hexylthiophene) (P3HT) of estimated molecular weight of 40 kDa and ICxA (a low cost fullerene mixture of mono-, bis-, and tris- adducted indene C$_{60}$ consisting of 53% indene-C60 bisadduct (ICBA), 36% indene-C60 monoadduct (ICMA), and 11% indene-C60-tris-adduct (ICTA) were synthesized in house at the Centre for Organic Electronics (COE), University of Newcastle, Newcastle, Australia, following reported procedures [15]. Zinc oxide ZnO nanoparticles were prepared consistent with the litera-

ture [28]. Isopropyl alcohol (IPA) and chlorobenzene were supplied from Sigma Aldrich, St. Louis, MO, USA.

## 2.2. Device Fabrication

Two standard geometry device types were fabricated with P3HT:ICxA as bulk heterojunction active layers in a configuration of Glass/ITO/PEDOT:PSS/BHJ active layer/ZnO or Ca/Al electron transport layer as shown in Figure 1a.

ITO-glass substrates were cleaned by rinsing in a sequence of solvents, milli-Q water, IPA, and acetone with 5 min of sonication for each solvent, then dried under a compressed nitrogen stream, and treated with UV Ozone for 15 min. Filtered PEDOT:PSS (PVP Al 4083, Clevios) solution was spin-coated onto the ITO substrates in ambient atmosphere with a spin rate of 4000 rpm (revolutions per minute) for 1 min and dried at 140 °C for 30 min, to form a uniform film $40 \pm 2$ nm thick of hole transport layer (HTL) material. The PEDOT:PSS-coated ITO substrates were transferred into a glovebox with less than 5 ppm moisture and approximately 5 ppm oxygen for active layer deposition. The active layer materials (P3HT and ICxA) were mixed in powder form and subsequently dissolved by sonication for 30 min and stirring at 60 °C for 1 h in 1 mL of chlorobenzene at a complete concentration of 35 mg·mL$^{-1}$ and ratio of donor and acceptor material of (1:0.8).

The P3HT:ICxA BHJ layer was spin-coated onto the PEDOT:PSS ITO substrates with a spin rate of 2000 rpm for 1 min to form a $100 \pm 5$ nm film, and dried at 110 °C for 5 min. For the P3HT:ICxA/ZnO device, ZnO nanoparticles with a concentration of 10 mg·mL$^{-1}$ were prepared as reported previously [28] as a suspension in isopropyl alcohol (IPA). The interlayer of ZnO nanoparticles was spin-coated onto the BHJ layer with a spin rate of 5000 rpm for 1 min and pre-annealed at 140 °C for 5 min to yield the $20 \pm 3$ nm thickness of the ZnO ETL. The thermal deposition of 100 nm thick Al electrodes was performed under vacuum ($2 \times 10^{-6}$ Torr). The P3HT:ICxA/Ca device was completed by evaporating a 30 nm calcium interlayer and a 100 nm aluminum cathode onto the active layer under vacuum ($2 \times 10^{-6}$ Torr). The actual P3HT:ICxA complete organic photovoltaic device (thickness $100 \pm 5$ nm) prepared on XY15S ITO-glass substrate is shown in Figure 1b.

## 2.3. Device Encapsulation

Encapsulation of organic photovoltaic devices isolates the active layer and electrodes from external $O_2$ and $H_2O$ [29] improving operational lifetime, with a requirement that the encapsulant should be chemically non-corrosive and electrically and thermally stable. The OPV devices were encapsulated inside a dry nitrogen atmosphere glove box, where a custom cut 1.1 mm thick cover glass, was bonded to the cell using a UV-curing adhesive (Lens Bond, Emsdiasum, Hatfield, UK). Although this approach seals the OPV device completely, the contact electrodes were not covered by the glass cover, allowing wires to be soldered to the two OPV device contacts (ITO and aluminum) using an USS-9210 Ultrasonic Soldering System (MBR Electronics, Wald, Switzerland) operated at 10 W and 60 kHz ultrasonic frequency. The fully encapsulated P3HT:ICxA device with wires soldered to the ITO and aluminum contacts is shown in Figure 1c.

## 2.4. Device Characterization

Current density–voltage (J–V) measurements and the device lifetime study were performed using the Class AAA calibrated LED solar simulator built at the Centre for Organic Electronics [30–33]. The LED solar simulator has a 360 mm × 240 mm test plane area which meets the flux requirement [34] with a 99.9% spectral match, 2% spatial nonuniformity of irradiance, 0.05% temporal instability of irradiance, and mismatch factor of 1%, over the 300–800 nm organic solar cell absorption spectrum. The LED solar simulator light intensity was calibrated to 100 mW/cm$^2$ using a FHG-ISG silicon reference photodiode (Infinite PV) [32].

### 2.4.1. J–V Measurement to Monitor the Device Degradation

The degradation testing protocols for devices defined by the International Summit on OPV Stability (ISOS) [35] were used for the J–V testing. Automated lifetime measurements of the OPV devices employed a 16-channel multiplexer with a Keithley 2400 source meter (Solon, OH, USA) to measure the J–V electrical parameters. Two degradation protocols have been followed: The ISOS–$L_2$ for measurement under light conditions required illumination of one sunlight at 100 mW/cm$^2$ irradiance, 65 °C temperature, and ~5% relative humidity for 18 days of continuous soaking light exposure. The ISOS–$D_2$ for dark measurements were 65 °C temperature and ~5% relative humidity for 18 days of dark storage.

### 2.4.2. EQE Measurement

External quantum efficiency (EQE) was measured over the wavelength range 300–800 nm in increments of 2 nm using a 100 W quartz tungsten lamp passed through an Oriel Cornerstone 130 monochromator (Newport, Irvine, CA, USA).

### 2.4.3. PL Spectroscopy

Photoluminescence spectroscopy (PL) was used to analyze the charge recombination behavior in devices. The PL spectroscopy of the encapsulated complete device layer stack was measured using a spectrofluorophotometer (Shimadzu RF–6000, Tokyo, Japan) to acquire the photoluminescence spectroscopy measurements at an excitation wavelength of 550 nm and scan speed of 200 nm/min. The bandwidths for excitation and emission were 5 nm and 10 nm, respectively. A cut-off filter (L–42) in the excitation light path removed wavelengths below 400 nm. All the spectroscopy measurements for the encapsulated OPV devices were recorded at room temperature in air. This allowed measurement of photo-bleaching light-induced degradation.

## 3. Results and Discussion

Six OPV devices were prepared for each electron transport layer (ETL) type to ensure quantitative measurements of J–V for both ETLs under ISOS–2 protocol light and dark degradation conditions. The average efficiencies of the devices were ~1.8% for the Ca ETL and ~3% for the ZnO ETL.

### 3.1. Effect of Light and Heat on Device Degradation Rate

Light illumination is a critical extrinsic degradation factor for OPV devices as shown in Figure 2a,b. There is a clear difference in the performance of the Ca and ZnO devices with both exhibiting degradation during the experiment; however, the Ca-based OPV device showed extremely rapid degradation in Figure 2a, as evidenced by the reduction in short-circuit current density ($J_{SC}$) and fill factor (FF).

This rapid degradation is consistent with the formation of an insulating calcium oxide layer at the active layer/Ca/Al interfaces [36], effectively blocking electron extraction, and introducing the "S-shape" feature to the $J_{SC}$ [37]. This behavior was not measured when ZnO was used as the ETL, Figure 2b, which strongly suggests that the degradation is due to an interaction between the Ca ETL and both the active layer and the aluminum top electrode. As illustrated in Figure 2c, the same degradation pathway is exhibited for the Ca device under thermal aging degradation conditions. The decay of the power conversion efficiency (PCE) in the Ca devices is most likely due to the formation of a calcium oxide layer. Both Ca and ZnO devices revealed a drop in the open circuit voltage ($V_{OC}$), related partially to operation at 65 °C. The electrical performance parameters of fresh and degraded OPV devices for each ETL under light exposure ISOS–$L_2$ and thermal dark ISOS–$D_2$ degradation protocols are shown in Table 1.

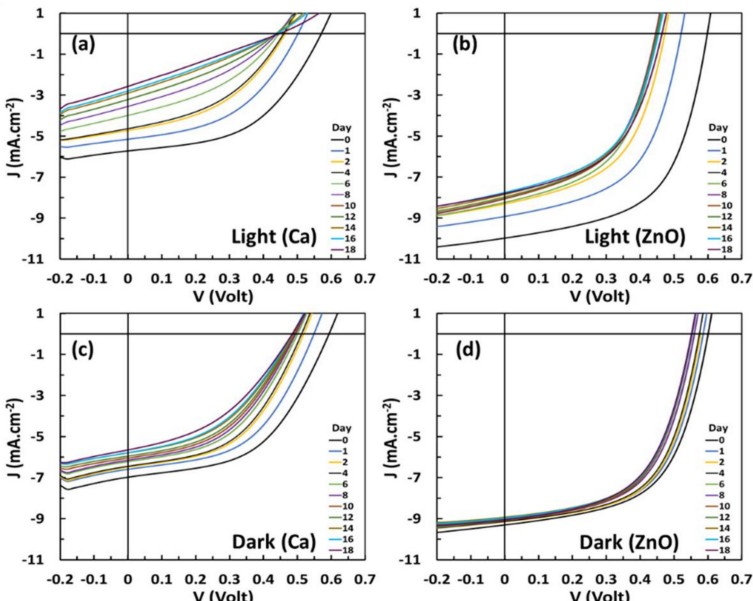

**Figure 2.** (**a**) J–V curves of the typical OPV device obtained for P3HT:ICxA with a Ca ETL layer under light condition (**c**) under dark condition; (**b**) J–V curves of the typical OPV device obtained for P3HT:ICxA with a ZnO ETL layer under light condition (**d**) under dark condition. The J–V characteristics are measured by applying linear sweep voltage to the photovoltaic device and evaluating the current using Keithley 2400 source meter. The simulator light intensity has been adjusted to 100 mW/cm$^2$.

**Table 1.** The electrical performance characteristic parameters of fresh and degraded OPV devices for the Ca or ZnO ETL layer under light as (ISOS−L$_2$) and dark (ISOS−D$_2$) protocols. Short circuit current (I$_{SC}$), open circuit voltage V$_{OC}$, fill factor (FF), and power conversion efficiency (PCE) are extracted from J–V measurements and these have been used to provide fundamental information about the behavior and characteristics of the photovoltaic device (Average ± Standard deviation (Best)).

| ISOS Protocol | ETL Type | PCE (%) | V$_{OC}$ (V) | FF | J$_{SC}$ (mA/cm$^2$) |
|---|---|---|---|---|---|
| Light (ISOS–L$_2$) | Fresh (ZnO) | 3.4 ± 0.04 (3.63) | 0.598 ± 0.01 (0.67) | 0.569 ± 0.02 (0.64) | 9.98 ± 0.27 (8.44) |
| | Degraded (ZnO) | 1.82 ± 0.16 (2.03) | 0.464 ± 0.03 (0.473) | 0.503 ± 0.023 (0.516) | 7.81 ± 0.47 (8.31) |
| | Fresh (Ca) | 1.62 ± 0.10 (2.4) | 0.571 ± 0.01 (0.70) | 0.495 ± 0.02 (0.55) | 5.72 ± 0.51 (6.37) |
| | Degraded (Ca) | 0.297 ± 0.0.8 (0.688) | 0.447 ± 0.04 (0.446) | 0.258 ± 0.023 (0.386) | 2.57 ± 0.7 (3.99) |
| Dark (ISOS–D$_2$) | Fresh (ZnO) | 3.21 ± 0.14 (3.44) | 0.601 ± 0.01 (0.66) | 0.574 ± 0.01 (0.66) | 9.30 ± 0.25 (9.94) |
| | Degraded (ZnO) | 2.77 ± 0.11 (3.04) | 0.551 ± 0.02 (0.575) | 0.554 ± 0.01 (0.578) | 9.05 ± 0.025 (9.14) |
| | Fresh (Ca) | 2.1 ± 0.23 (2.27) | 0.594 ± 0.01 (0.69) | 0.507 ± 0.02 (0.54) | 6.98 ± 0.45 (6.96) |
| | Degraded (Ca) | 1.12 ± 0.48 (1.47) | 0.485 ± 0.01 (0.495) | 0.409 ± 0.02 (0.462) | 5.65 ± 2.4 (5.14) |

To examine the detailed trends associated with ETL type, all the OPV device photovoltaic parameters extracted from the J–V curves in Figure 2 are plotted as a function of light exposure time in Figure 3. The Ca and ZnO devices seem to both display a two-regime degradation behavior with the initial loss an exponential degradation followed by a period

of linear degradation. The PCE of the Ca devices decreased by 55% and the ZnO devices by 40% over the first 72 h of operation. These losses in Ca devices resulted from a 33% reduction in $J_{SC}$, a 20% reduction in FF, and a 20% reduction in $V_{OC}$, that almost stabilized after the burn-in period. However, the PCE of Ca devices continued to rapidly decrease to 8% of PCE initial. In remarkable contrast, the ZnO ETL-based OPV devices demonstrated much better long-term stability with the efficiency loss ascribed to 15% from $J_{SC}$, 21% from $V_{OC}$, and 10% in FF which almost stabilized after the burn-in period, following which the PCE slowly decreased to approximately 50% PCE initial.

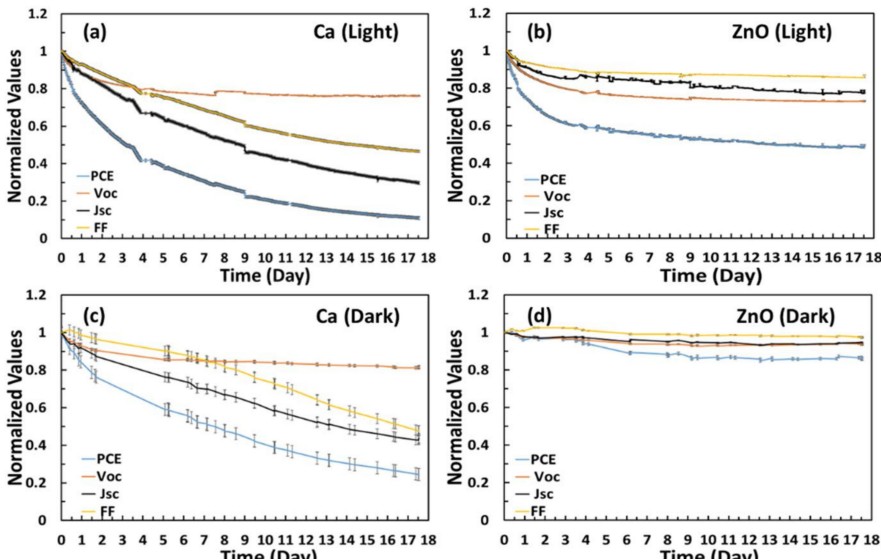

**Figure 3.** (**a**) Temporal stability of device parameters of power conversion efficiency (PCE), open circuit voltage ($V_{OC}$), short circuit current density ($J_{SC}$), and device fill factor (FF) as a function of light exposure time for both Ca ETL under light ISOS–$L_2$ and (**c**) dark ISOS–$D_2$ degradation conditions, (**b**) for ZnO ETL under light (**d**) dark.

All parameters have been normalized to their initial value upon commencing the stability testing.

This result represents a significant stability difference between OPV/ZnO devices and OPV/Ca devices, which is further highlighted under dark degradation conditions (ISOS–$D_2$) of 65 °C and ~5% relative humidity where the Ca device, Figure 3c, degraded rapidly while the ZnO device, Figure 3d, has a very small change in PCE, with this being related to the drop in the $V_{OC}$ value due to thermal effects.

This contrast identifies elevated light illumination as a major degradation factor of the device, rather than the active layer material itself, or the thermal degradation, which provides a very valuable tool to isolate the degradation processes in differently structured OPV devices. In conclusion, the ZnO ETL devices demonstrate excellent electron transport properties and a much longer lifetime than Ca devices.

### 3.2. Scanned Laser Induced Current

A laser beam-induced current (LBIC) method, where a tightly focused laser is scanned over the photovoltaic cell, provided the photocurrent measurements for both fresh and degraded P3HT:ICxA (Ca and ZnO) devices. The photocurrent map homogeneity is visible as variations in contrast between dark and bright regions, with a region of high photocurrent density being bright white.

Images of fresh and degraded devices formed from the LBIC data are shown in Figure 4 where the top panels are from fresh devices, whereas the bottom images show degraded devices. Eighteen days of irradiation under test conditions caused photobleaching and

oxidization which are clearly visible on the degraded Ca device as dark regions introduced in the photocurrent map.

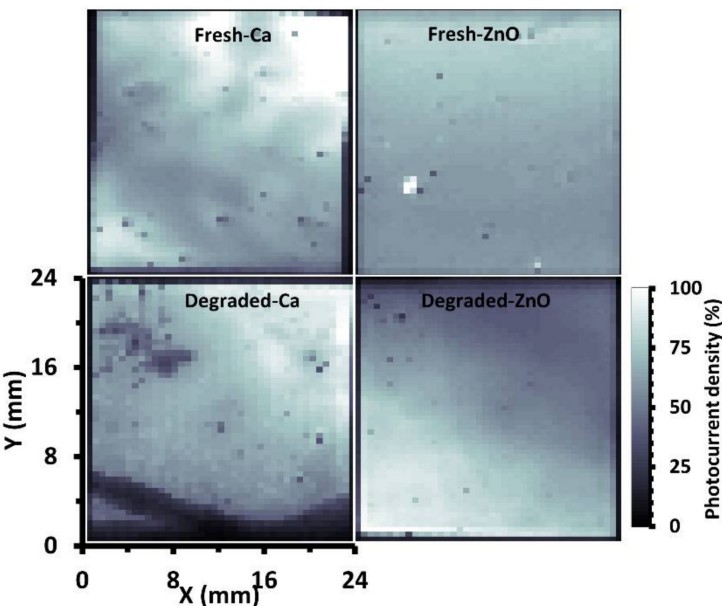

**Figure 4.** LBIC images of the fresh and degraded P3HT:ICxA/Ca and P3HT:ICxA/ZnO devices measured after 18 days of illumination under one sunlight at 65 °C. The scale bar is a normalized value of the photocurrent density for each device.

These changes are not from failure of the device encapsulation, but reflect the change in Ca ETL performance with time, evident as discrete dark zones. This degradation is not evident in the ZnO ETL device after the same ISOS-L$_2$ testing regime.

### 3.3. External Quantum Efficiency (EQE)

The comparison of EQE spectra of fresh and degraded devices is presented in Figure 5. EQE significantly decreased over the entire spectral range (300–800 nm) after 18 days of degradation for both light and dark conditions of Ca devices. Figure 5a illustrates the Ca device EQE exhibiting a ~50% reduction of spectral response without substantial variation in spectral shape.

Based on the device light soaking, the general shape of the EQE curves only changes slightly, indicating that the rate of active layer degradation is slight. The same drop in EQE value is recorded for both light and dark degraded Ca devices, Figure 5a,c. Alternatively, the EQE spectrum of ZnO devices revealed a minor change in EQE for the dark degraded device after 18 days of aging; however, there was a remarkable change in EQE device efficiency ~45% for the ZnO light degraded devices as shown Figure 5b. This drop in device efficiency is related to the burn-in mechanism that changes the morphology of the active layer material of the OPV device. In extraordinary contrast, as demonstrated in Figure 5d, the EQE measured for the ZnO device of the dark conditions reveals a long-term stability with only a ~1% reduction of spectral response without substantial variation in spectral shape.

The EQE results for both Ca and ZnO dark devices show a significant difference between the Ca and ZnO devices. The loss in device efficiency of both Ca and ZnO devices after 18 days of light and dark degradation is about the same rate, 50%. The J$_{SC}$ value arising from integrating the EQE distribution for the Ca device does not match the measured J$_{SC}$, with a mismatch value more than 40%, confirming the slight light soaking effect improvement in J$_{SC}$ for the Ca device during the first hour of continuous light illumination. In contrast, the mismatch value for J$_{SC}$ of a ZnO device is less than 8%. Moreover, the almost perfect match of EQE for ZnO device in the dark before and after

degradation shown in Figure 5d, in sharp contrast to the EQE drop after degradation under the light (Figure 5b), shows a significant light induced drop in photocurrent, likely due to charge recombination at the interface or photo-bleach in the active layer, which is consistent with the earlier observation in Figure 3b,d. This result indicated photo-induced device deterioration with our current active materials is still unstoppable even though the device has been encapsulated. The promising side of this study is that using ZnO ETL, the device still retained 40% of the initial PCE.

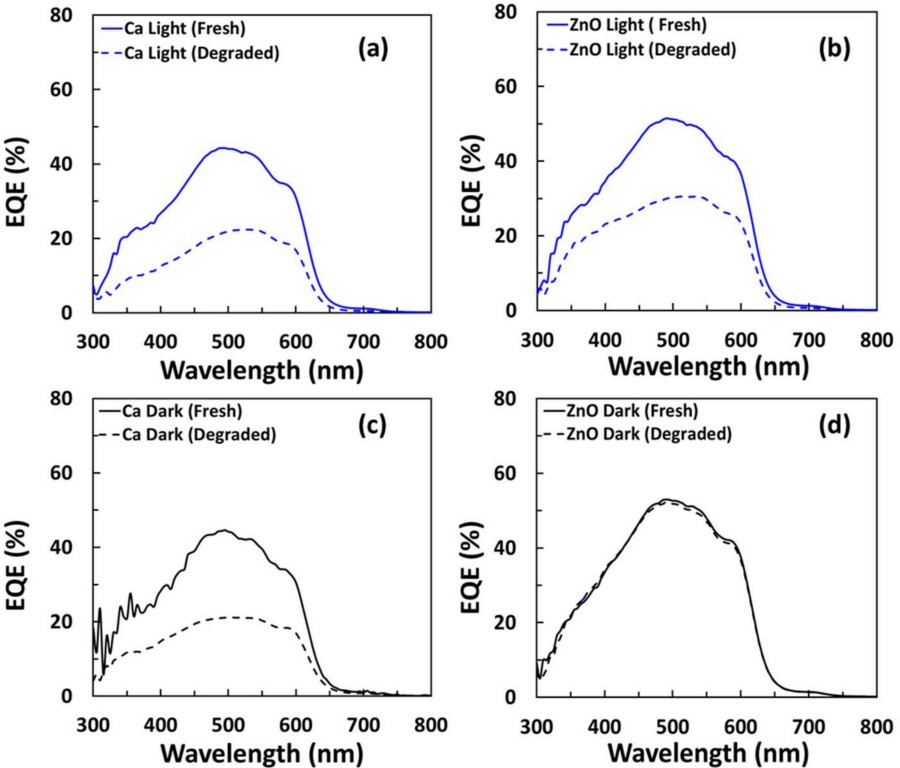

**Figure 5.** The External Quantum Efficiency (EQE) for Ca (**a**,**c**) and ZnO (**b**,**d**) OPV devices, solid lines denote fresh devices and dashed lines represent degraded OPV devices (blue color means light and black color means dark conditions). The EQE characteristics are measured within 2 °C of the reference temperature (25 °C). The scan recorded data at a 2 nm interval over the wavelength range 300–800 nm.

### 3.4. Photoluminescence (PL) Spectra

Photoluminescence (PL) properties are highly sensitive to material local structure and spatial distribution and concentrations of defects [38]. A lower PL intensity is desirable as PV operations require charge extraction rather than recombination and photon emission. Generally, photoluminescence intensity signal changes can be caused by three factors: exciton quenching at the organic–metal interface [39], exciton quenching at the donor–acceptor boundary, and changes in the optical absorption distribution [36,40].

Photoluminescence measurements on the P3HT:ICxA/ZnO and P3HT:ICxA/Ca OPV devices provide further evidence for the formation of CaO. In practice, the P3HT:ICxA OPV devices display two emission peaks at 660 and 720 nm. The PL spectra for the P3HT:ICxA/Ca devices show a significant PL intensity at 660 nm under both light and dark degradation, where the maximum values of fresh devices are a factor of 2.3 higher than for a degraded device under light illumination and 1.4 times higher than for a degraded Ca device under dark conditions. These differences indicate the reduction in photo-induced electron transfer from the active layer after degradation. Because of the CaO build-up, some electrons cannot reach the ETL and recombine radiatively leading to the increase in photoluminescence signal intensity [41]. This enhancement in PL of Ca devices indicates the

reduction of the interface area between the active layer and the ETL where PL quenching occurs [42]. This phase separation reduces charge extraction, which decreases charge carrier transport in both donor and acceptor phases after degradation [43,44]. As the absorption of the active material (P3HT:ICxA) did not change upon degradation, as seen in Figure 6, we conclude that the change of the PL intensity initiates from CaO formation between the active layer and Ca ETL.

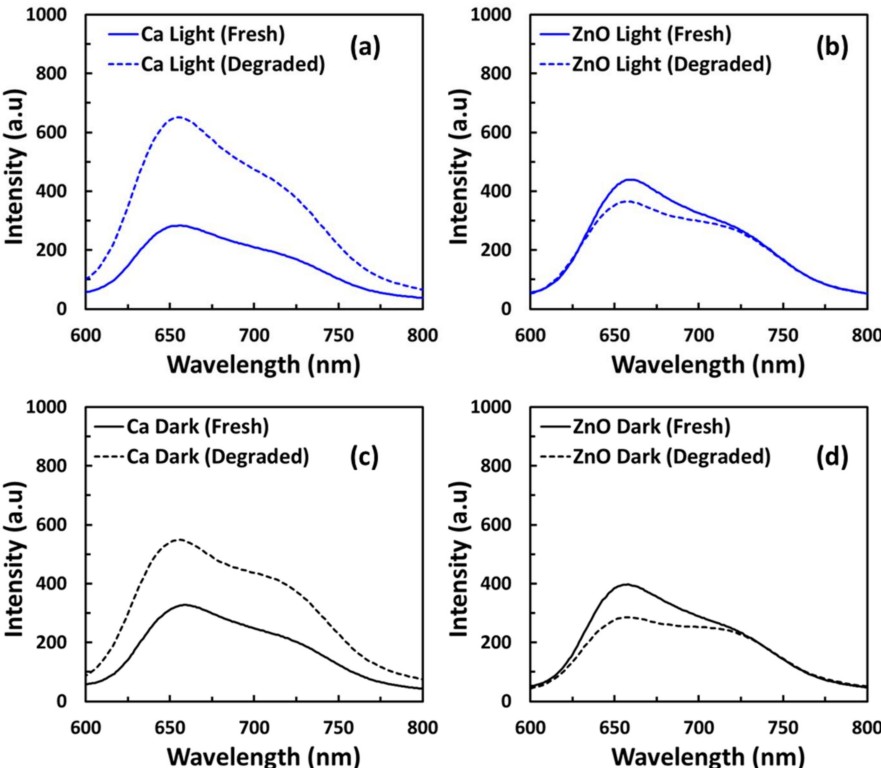

**Figure 6.** (**a**,**c**) Photoluminescence spectra of P3HT:ICxA/Ca, and (**b**,**d**) Photoluminescence spectra of P3HT:ICxA/Ca systems. Solid lines denote fresh devices and dashed line represents degraded devices (blue color means light and black color means dark conditions). The photoluminescence spectra are measured within 2 °C of the reference temperature (25 °C). A 550 nm excitation wavelength was used with luminescence recorded at 200 nm/min scan speed with 1 nm data interval and a 65° light incident angle.

The degradation of Ca ETL creates CaO that prevents the active layer from having efficient contact with the Ca ETL. These results are consistent with the drop in device PCE for Ca ETL OPV degraded devices, whereas P3HT:ICxA/ZnO devices show an inverse scenario, as PL intensity decreases at 660 nm with around 17% for ZnO device under light and ~27.5% for ZnO device in the dark. The reduction in PL intensity after degradation is related to non-radiative recombination mechanisms [45] that are caused by the increase in trapped states preferentially capturing a single carrier type [45,46], which reduces the current density in the OPV devices, negatively impacting the OPV device performance.

## 4. Conclusions

For the encapsulated OPV devices, if the up-scale roll-to roll printable production is the future, understanding and controlling each layer in the device structure is essential to guarantee a usable device. This study has examined one element in the device architecture at the electron transport layer and verified that light and thermal degradation of OPV devices with two different ETLs (Ca and ZnO) is not only due to changing the morphology of the active layer. Depending on the material used on the ETL, the kinetics of degradation vary considerably. In particular, OPV devices using Ca as ETL exhibit a strong and rapid

drop in PCE and in the LBIC photocurrent density images which are not observed in ZnO based OPV devices. This interface degradation mechanism is associated with a decay in $J_{SC}$ and FF which may be explained by the formation of an isolating calcium oxide layer at the active layer/Ca/Al interfaces, which decreases the electron extraction. The poor performance of Ca ETL OPV-based devices with thermal dark aging showed similar results as the light degradation scenario. Additionally, physical analysis using PL spectra showed an increase in PL signal for devices with Ca as ETL, which is explained by a reduced charge carrier extraction efficiency and more electrons recombining radiatively to cause an increase in photoluminescence signal intensity. The comparison between the Ca and ZnO ETL layers in this study finds the ZnO ETL devices to be more stable and efficient over time than the Ca ETL devices. Our work illustrated that ZnO can potentially be a good candidate as the electron transport layer for printable OPV devices.

**Author Contributions:** Conceptualization and supervision, B.V., J.H., X.Z., W.B. and P.D.; investigation, A.A.-A.; resources, COE and ANFF Facility; writing—original draft preparation, A.A.-A.; writing—review and editing, A.A.-A., B.V., J.H., X.Z., W.B. and P.D. All authors have read and agreed to the published version of the manuscript.

**Funding:** This work was supported by the University of Newcastle through the Centre for Organic Electronics.

**Institutional Review Board Statement:** Not applicable.

**Informed Consent Statement:** Not applicable.

**Data Availability Statement:** Not applicable.

**Acknowledgments:** The work was also performed in part at the Material Node of the Australian National Fabrication Facility under the National Collaborative Research Infrastructure Strategy to provide nanofabrication and microfabrication facilities for Australia's researchers.

**Conflicts of Interest:** The authors declare no conflict of interest.

## Abbreviations

| | |
|---|---|
| ITO | Indium tin oxide |
| PVP | Polyvinylpyrrolidone |
| ETL | Electron transport layer |
| HTL | Hole transport layer |
| Ca | Calcium |
| CaO | Calcium oxide |
| ZnO | Zinc oxide |
| PV | photovoltaic |
| OPV | Organic photovoltaic |
| ISOS | International Summit on OPV Stability |
| ISOS–$L_2$ | light ISOS–$L_2$ degradation conditions (irradiance = 100 mW/cm$^2$, temperature 65 °C, and relative humidity) |
| ISOS–$D_2$ | dark ISOS–$D_2$ degradation conditions (irradiance = 0 mW/cm$^2$, temperature 65 °C, and relative humidity) |
| PL | Photoluminescence |
| J–V | Current density–voltage |
| BHJ | Bulk heterojunction |
| PCE | Power conversion efficiency |
| $I_{SC}$ | Short circuit current |
| $V_{OC}$ | Open circuit voltage |
| FF | Fill factor |
| EQE | External quantum efficiency |
| IEA | International Energy Agency |
| P3HT | Poly(3-hexylthiophene) |

| ICxA | Low-cost fullerene mixture of mono-, bis-, and tris- adducted indene C60 consisting of 53% ICBA, 36% ICMA, and 11% ICTA |
| --- | --- |
| ICBA | 1′,1″,4′,4″-tetrahydro-di[1,4]methanonaphthaleno[5,6]fullerene-C60 |
| ICMA | 1′,4′-Dihydro-naphtho[2′,3′:1,2][5,6]fullerene-C60 |
| PEDOT:PSS | Poly(3,4-ethylene dioxythiophene) polystyrene sulfonate |
| ITO | Indium tin oxide coated glass |
| IPA | Isopropyl alcohol |
| LED | Light-emitting diode |
| LBIC | Laser beam induced current |

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
