# Peer review of "The Role of the Electron Transport Layer in the Degradation of Organic Photovoltaic Cells"

_coatings, doi:10.3390/coatings12081071_

Round 1

Reviewer 1 Report

The authors investigated the effects of light and thermal degradation of OPV devices with two different ETLs (Ca and ZnO), reporting a good characterization of the layers used and the devices fabricate.

Why did the authors decide to fabricate standard configuration devices? Given that those fabricate in the inverse configuration already have a greater temporal stability of the efficiencies and therefore a greater resistance to degradative effects? The authors must justify the choice of the standard configuration and the type of materials used in their study, unfortunately the introduction lacks motivation. The design and fabricate of the devices is not focused.

Author Response

We thank the reviewers for their constructive comments. Based on their comments, we have made the following revisions.

Reviewer 2 Report

Please address the following comments point by point.

In the introduction section, it is important to discuss some previous studies related to degradation mechanisms because many studies show how light and temperature influence the device's performance. By relating their work, please provide a comparative analysis to strengthen your case. At the moment, it is difficult to understand that what is the significance of the proposed work?

In the experimental section, please provide complete information about the device fabrication. For example, please provide the spin speed, time, temperature, etc of all the deposited layers.

The OPVs have been examined by indoor light sources, though the indoor environment is less harsh compared to 1-sun. Please study and include the following recent important studies related to indoor OPVs in the introduction section. Also, how this experiment will be related to indoor characterizations? Which layer may have a significant effect under these conditions? Please comment. https://doi.org/10.1002/adfm.202201921; https://doi.org/10.1016/j.jpowsour.2021.230782; https://doi.org/10.1002/aenm.202003103;

There is a need to discuss the EQE drop in the dark. What factors are contributing to this drop? Also, it is suggested to include the maximum PCE values in Table 1.

Is there any specific reason for choosing ICxA material? Because there are many state-of-the-art materials available that possess high efficiencies. Referring to comment 1, there is a need to highlight the proposed work.

            Please use minutes or min, hours or hr, seconds or sec. There are some issues of consistency in using these terminologies.  

Author Response

(The authors gave the same response as above.)

Reviewer 3 Report

Dear authors, I have read your paper and I have seen a nice work behind. However, in my opinion, the paper is sometimes difficult to follow and more information is required on some issues. My comments:

-Clarify better the innovation and goal of this work in the abstract and in the main text.

-The Introduction chapter is short, this needs to be expanded.

-The manuscript must be at least 15-20 pages long. For this reason, it is necessary to present the literature in much more depth and to compare it with the current research.

-At the end of the first chapter, it is necessary to summarize what chapters will follow. In addition, in this section, the innovative novelty of this research should also be highlighted.

-The Results and Discussions chapter should be divided into a Results and a Discussions chapter, according to the MDPI specification.

-The Discussion section is missing.

-Please, avoid lumped references. What is the contribution of each reference?

-The conclusion chapter is short. Extend the conclusion with more general usability. What are the benefits of the results in a global context? Please explain this better in the manuscript.

-At the end of the study need to create an abbreviation table.

Author Response

(The authors gave the same response as above.)

Round 2

Reviewer 2 Report

Article can be accepted for adjusting the references corrections. Currently, it does not follow journal's guidelines.

Author Response

Dear editor,

As the reviewer suggested, we have revised our references accordingly, using the ACS style.

Best regards

Xiaojing